# *GCKR* rs780094 Polymorphism as A Genetic Variant Involved in Physical Exercise

**DOI:** 10.3390/genes10080570

**Published:** 2019-07-28

**Authors:** Isabel Espinosa-Salinas, Rocio de la Iglesia, Gonzalo Colmenarejo, Susana Molina, Guillermo Reglero, J. Alfredo Martinez, Viviana Loria-Kohen, Ana Ramirez de Molina

**Affiliations:** 1Nutrition and Clinical Trials Unit, IMDEA Food CEI UAM + CSIC, 28049 Madrid, Spain; 2Department of Pharmaceutical and Health Sciences, Faculty of Pharmacy Universidad San Pablo-CEU, CEU Universities, Urbanización Montepríncipe, 28925 Madrid, Spain; 3Biostatistics and Bioinformatics Unit, IMDEA Food CEI UAM + CSIC, 28049 Madrid, Spain; 4Department of Production and Characterization of New Foods, Institute of Food Science Research (CIAL) CEI UAM + CSIC, 28049 Madrid, Spain; 5Department of Food Sciences and Physiology, University of Navarra, 31009 Pamplona, Spain; 6CIBERobn, Instituto Carlos III, 28029 Madrid, Spain

**Keywords:** glucokinase-regulator, exercise, behavior, genotyping, obesity

## Abstract

Exercise performance is influenced by genetics. However, there is a lack of knowledge about the role played by genetic variability in the frequency of physical exercise practice. The objective was to identify genetic variants that modulate the commitment of people to perform physical exercise and to detect those subjects with a lower frequency practice. A total of 451 subjects were genotyped for 64 genetic variants related to inflammation, circadian rhythms, vascular function as well as energy, lipid and carbohydrate metabolism. Physical exercise frequency question and a Minnesota Leisure Time Physical Activity Questionnaire (MLTPAQ) were used to qualitatively and quantitatively measure the average amount of physical exercise. Dietary intake and energy expenditure due to physical activity were also studied. Differences between genotypes were analyzed using linear and logistic models adjusted for Bonferroni. A significant association between *GCKR* rs780094 and the times the individuals performed physical exercise was observed (*p* = 0.004). The carriers of the minor allele showed a greater frequency of physical exercise in comparison to the major homozygous genotype carriers (OR: 1.86, 95% CI: 1.36–2.56). The analysis of the *GCKR* rs780094 variant suggests a possible association with the subjects that present lower frequency of physical exercise. Nevertheless, future studies are needed to confirm these findings.

## 1. Introduction

Physical exercise and energy intake are fundamental factors in the energy balance equation and in body weight management [1]. Knowledge about the factors associated with physical activity practice and energy expenditure as well as the potential mechanisms involved in fuel homeostasis is essential in order to understand human thermodynamics [1]. 

The energy used during physical performance is the most variable component of total energy expenditure [2]. This feature depends on different factors such as body composition, intensity and duration of the physical activity, and the individual genetic profile [3,4]. Also, the sport aptitude has a strong genetic component [5,6]. However, there is a lack of research about the role that genetic variability plays in the motivation for practicing any physical exercise, the adherence to it and the related efficiency.

Multiple metabolic and physiological processes have been related to physical training [7]. The study of genetic variants occurring in genes associated with the regulation of such metabolic processes could help determine how these processes are implicated in the practice of physical exercise. Although there are Genome-wide association studies (GWAS) linking specific locus with exercise and, also, reviews determining associations between SNPs with sports performance, there is scarce evidence on the relationship between obesity and metabolism-associated polymorphisms with physical exercise [8,9,10,11]. Further studies may be of interest to establish more precise relationships between genes related to metabolism and how they affect the practice of physical exercise. 

Thus, genes related to inflammation such as those encoding interleukins, or the C-reactive protein have been linked to athletic performance [12]. In turn, genes associated with energy metabolism such as *FTO* and *POMC*, as well as with circadian rhythms like *CLOCK* and *PER2* gene, have been related to the muscular system or the response to sport and exercise [13,14,15,16]. In the case of vascular function, genes such as the *NOS3* gene, whose effect is modified by the practice of physical exercise, or the *GNB3* gene, known to be associated with elite athletes, have also been investigated [17,18]. Moreover, studies of genetic variants involved in lipid metabolism such as apolipoproteins or peroxisome proliferator-activated receptors (*PPARs*) and in carbohydrate metabolism such as *ADIPOQ*, have been associated with sports practice, which could also influence predisposition to the practice of physical exercise [19,20,21].

In this context, the study of the glucokinase regulatory protein (*GCKR*) gene, involved in lipid and carbohydrate metabolism, can be of interest as it might also interfere with the amount of physical exercise performed [22]. The *GCKR* gene, located on chromosomal region 2p23.2–3, modulates glucokinase (GCK) that is a key regulatory enzyme of glucose metabolism and storage, and potentially implicated in energy utilization [23,24]. Thus, a hypothetical mechanism behind a physically active or sedentary behaviour may lie on the greater or lesser availability of energy substrates, as well as the signals these substrates exert at the brain level.

Recently, the single nucleotide polymorphism rs780094, located in an intronic region of the *GCKR* gene, was found to be related with triacylglycerides (TAG) and fasting plasma glucose concentrations [23]. Specifically, the major allele (C) of *GCKR* rs7800094 is associated with decreased TAG, but increased fasting plasma glucose, while minor allele (T) is associated with lower levels of fasting plasma glucose and insulin, and higher levels of TAG [25]. In this regard, it is interesting to point out that GCKR has been identified in the same brain area than GCK, what could modulate the feeding behavior and energy balance [26]. GCKR and GCK, which act as glucose-level sensors, might interact with appetite-regulating peptides and interfere in the individual feeling of satiety [27]. 

Information about genetic variants that modulate energy intake and expenditure through the adherence to physical exercise performance can be useful to detect those subjects with lower frequency of exercise practice and those less prone to maintain an adequate energy balance to maintain a healthy status, to prevent overweight and obesity, and to personalize loss weight strategies. In this context, the aim of the present study was to identify genetic variants that modulate the amount of physical activity and exercise performed to detect those subjects that exercise less, with the goal to prevent overweight.

## 2. Materials and Methods 

### 2.1. Subjects and Study Protocol

A total of 557 subjects (155 men and 402 women, aged from 18 to 65 years), who participated in a GENYAL Platform (Platform for Clinical Trials in Nutrition and Health) observational study, encompassed this investigation. Participants were recruited in Cantoblanco Campus (Autonomous University of Madrid, Madrid, Spain), being almost entirely of Caucasian origin. Inclusion criteria were: free-living adults aged from 18 to 70 years that gave written informed consent to be contacted to perform clinical trials and nutritional intervention studies. Exclusion criteria were: To suffer from any serious illness (kidney or liver diseases or other condition that affects lifestyle or diet), to present dementia or impaired cognitive function and to be pregnant or breastfeeding. Of the total of the participants, it was not possible to obtain all the data. Thus, only 490 were evaluated for physical activity and exercise practice while only 451 were genotyped. The study was conducted according to the guidelines laid down in the Declaration of Helsinki, and all procedures involving human subjects were approved by the Research Ethic Committee of Autonomous University of Madrid (CEI 27-666). Written informed consent was obtained from all subjects. 

### 2.2. Anthropometric Measures and Dietary Intake

Anthropometric and body composition variables, such as height, weight, fat mass percentage and waist circumference, were measured by standard validated techniques [28]. Body weight and fat mass percentage were assessed using the body composition monitor BF511 (Omron Healthcare UK Ltd., Kyoto, Japan). BMI was calculated as the body weight divided by the squared height (kg/m^2^). Waist and hip circumferences were measured using a Seca 201 non-elastic tape (Quirumed, Valencia, Spain). A 72-h food record was collected from all participants. DIAL (2.16 version, Alce Ingeniería, Las Rozas, Madrid, Spain) Software was used to analyze the energy intake, macro and micronutrients of the collected records [29]. 

### 2.3. Physical Activity Measures

The frequency of physical exercise practice on a regular basis (“exercise” variable), was quantified by a specific question in order to define how many times per week they used to practice physical exercise: 0, 1, 2, 3, 4 or 5 and > 5 times. A dichotomized version of this variable was created (“exercise classification”) as follows: (1) Subjects who practiced physical exercise 0 times per week and (2) subjects who practiced physical exercise at least one day per week. This classification was made in order to better split the sample in terms of exercise practice: those who did not practice at all vs. those who practiced some exercise.

In addition, Minnesota Leisure Time Physical Activity Questionnaire (MLTPAQ) was used to quantitatively measure the average physical activity practice (kcal/day) by the volunteers [30]. Based on the Compendium of physical activities, Energy Expenditure in Physical Activity (EEPA) was estimated as follows: EEPA = I × N × T; where “I” represents the degree of intensity for each physical activity in kilocalories / minute; “N”, the number of times that physical activity was developed; and “T”, the time in minutes spent in each session [31].

### 2.4. Biochemical Measurements

Blood samples were taken early in the morning at IMDEA Food after a 12-h overnight fast and stored at 4 °C to 6 °C until analysis (always performed within 48 h) by Laboratory CQS Consulting. This laboratory integrates preanalytical, analytical and post-analytical processes. Total cholesterol (TC), high-density lipoprotein (HDL) and low-density lipoprotein cholesterol (LDL), triacylglycerols (TAG) and glucose were determined by enzymatic spectrophotometric assay using an Architect CI8200 instrument (Abbott Laboratories, Chicago, IL, USA). The triglyceride-glucose index (TyG index) was calculated as the natural logarithm (Ln) of the product of plasma glucose and TAG according to the following formula: Ln (TAG (mg/dL) × glucose (mg/dL))/2 [32].

### 2.5. DNA Isolation and Genotyping 

A total of 64 genetic variants related to inflammation (7 genes; 9 SNPs), circadian rhythms (2 genes; 6 SNPs), vascular function (8 genes; 10 SNPs) as well as energy (11 genes; 17 SNPs, lipid (9 genes; 15 SNPs) and carbohydrate metabolism (3 genes; 7 SNPs), were analyzed. The selection was based on a previous study in accordance with the allelic frequency and TaqMan^®^ probe (Applied Biosystems, Waltham, MA USA) availability [22]. Blood samples were taken and stored at −80 °C until DNA extraction. Genomic DNA from each participant was isolated from whole blood using the QIAamp DNA Blood Mini Kit (Qiagen Sciences, Inc., Germantown, MD, USA) and recovered in 100 mL of nuclease-free water. Its concentration and quality were then measured in a nanodrop ND-2000 spectrophotometer (ThermoScientific, Waltham, MA, USA). The mean concentration of the samples was 80 to 90 ng/mL. Genotyping was performed using the QuantStudio_ 12 K Flex Real-Time PCR System (Life Technologies Inc., Carlsbad, CA, USA) with a TaqMan OpenArray plates following the manufactured instructions (Real-Time PCR Handbook and education center of Applied Biosystem) [33]. The results were analyzed using TaqMan Genotyper software (V 1.3, Applied Biosystems, Waltham, MA USA). The proportion of genotypes not passing the quality threshold was <5%.

### 2.6. Statistical Analysis

Statistical analysis was performed with the R software, version 3.4.1 (R Foundation for Statistical Computing, Vienna, Austria) [34]. Deviations from Hardy-Weinberg equilibrium of genotype frequencies at individual loci were assessed using standard χ2 tests. Descriptive analyses were implemented for different continuous and categorical variables by sex, SNP (single nucleotide polymorphism) rs780094 and exercise classification. *p*-Values were obtained using one-way ANOVA for continuous variables, and Fisher exact test for categorical variables. Associations between the two variables, exercise variable and energy expenditure due to physical activity per week, were assessed by computing the Pearson correlation coefficient (r) and the corresponding statistical test. Identification of significant SNPs was performed by deriving logistic model vs. SNPs, adjusted by sex, age and additional appropriate covariables when necessary. Model assumptions were checked in all the cases. Three logistic genetic models were evaluated: Additive, dominant and codominant. These genetic models refer to the dominant or recessive character of each of the alleles. Thus, the additive model indicates a multiplicative effect depending on the number of rare alleles. In the dominant model, the less frequent allele is considered dominant (common homozygous vs. heterozygous + rare homozygous), and the codominant model considers each genotype individually, with independent genetic effects. The Bonferroni method was used to correct the *p*-values for multiple test (64 SNPs). Significance level was set to α = 0.05 adjusted after Bonferroni correction. No power analysis could be conducted as there was no idea of the variability of the responses analyzed in advance, and the sample used was the largest available.

## 3. Results

Out of all the genetic variants studied, only one significant statistical association between *GCKR* rs780094 polymorphism and the frequency of physical activity performed was found, when adjusted for multiple corrections. Considering the number of polymorphisms studied and the obtained results, only the related ones to this association will be discussed.

Descriptive statistics concerning gender, exercise classification and *GCKR* rs780094 data showed several differences regarding population characteristics (Table 1). This table shows anthropometric measures, biochemical variables, dietary intake and physical activity and exercise characteristics, according to gender, exercise classification (inactive vs. active subjects, see above) and genotype *GCKR* rs780094. 

The analysis showed significant differences concerning gender for the total anthropometric variables. Thus, males presented significantly higher values of body weight, height, BMI, lean mass percentage, visceral fat, waist circumference and waist-hip ratio than females. On the other hand, fat mass in males resulted in significantly lower values than females. Several biochemical parameters were also significantly different concerning gender. Hence, women had higher total cholesterol and HDL cholesterol levels while triglycerides/glucose index, Total Cholesterol/HDL and LDL/HDL were of lower values. Besides, men consumed and expended statistically significant more calories, as well as performed more times of physical exercise per week than women. 

Concerning exercise classification (inactive subjects who never practiced any kind of physical exercise per week vs. active subjects who did once or more times per week), significant differences for several anthropometric variables were expected (Table 1). Indeed, BMI (*p* = 0.005), fat mass (*p* = 0.001), visceral fat classification (*p* = 0.049), and waist circumference (*p* = 0.042) were more elevated in inactive individuals as compared to active subjects. Moreover, water intake resulted in statistically significant differences according to exercise classification variable, where active subjects consumed more water compared to inactive subjects (*p* = 0.001). Likewise, energy expenditure due to physical practice (calories were calculated from the METs, **metabolic equivalents of task,** results for the physical activity practice registered in the MLTPAQ) resulted in statistically significantly differences in terms of exercise classification (*p* < 0.001). No significant differences were found for biochemical parameters.

The association between energy expenditure by physical activity and exercise variable, has a Pearson correlation of *r* = 0.37 (*p* = 8 × 10^−17^), indicating a significant, although moderate, association between these two variables. 

Concerning the genotype analyses, for *GCKR* rs7800094, genotype frequencies were in Hardy–Weinberg equilibrium (*p* = 0.116). No significant differences were found in anthropometric variables (Table 1), although there were significant differences regarding total cholesterol (*p* = 0.041). Total caloric value of fat intake resulted in statistically significant differences since rare homozygous carriers consumed more amount of fat in contrast to mayor allele carriers (*p* = 0.036). In this sense, a tendency for a different energy intake has been observed (*p* = 0.074), where rare homozygous carriers consumed more total energy than mayor allele carriers. At the same time, there were significant differences in exercise variable (Table 1). Particularly, rare homozygous carriers practiced physical exercise a greater amount of times per week compared to major allele carriers who presented lower physical exercise per week (*p* = 0.016). There were no significant differences with respect of energy expenditure attributed to physical activity practice concerning genotype (*p* = 0.582). 

In this study, the association between the above described 64 genetic variants and exercise classification, were tested by developing logistic regression models adjusted by sex, age and BMI. Of the total of the polymorphisms analyzed, only *GCKR* rs780094 was found to be significantly associated with exercise classification after correction for multiple tests. Thus, Table 2 shows the distribution of our sample with regards to rs780094 genotype and exercise classification. It displays the corresponding odds ratios and *p*-values of the three logistic models: Codominant, additive and dominant. Statistically significant differences were found for the three designs. Within the exercise classification variable: active subjects, there was a higher percentage of rare homozygous carriers in comparison with mayor homozygous carriers. Thus, Figure 1 shows the sample distribution of the exercise classification variable according to the genotype. Almost 40% of common homozygote carriers did not perform any physical exercise per week, compared to the homozygous variants where the percentage of no-active people was much lower (14%). Most heterozygous carriers (86%), performed at least some type of exercise once a week (*p* = 0.004).

## 4. Discussion

Physical exercise provides a plethora of health benefits, whereas sedentary lifestyle is considered to be a main risk factor in the development of multiple chronic diseases [35]. Currently, there is a scarcity of knowledge about the mechanisms that trigger the practice of physical exercise and energy expenditure. However, this knowledge in the area could lead to personalized strategies focused on people prone to weight gain [36], even before overweight manifests itself [22]. Therefore, research on the mechanisms that modulate energy expenditure could be useful to reduce the overweight rates of the population.

Genetic factors could play an influential role in the behavior of individuals regarding their physical exercise [5]. In this context, the *GCKR* gene which regulates glucokinase activity in the liver, influences the regulation of lipid metabolism and hepatic glucose, so that it is strongly associated with fasting TAG and glucose levels [37]. Specifically, it has been observed that the minor T allele of *GCKR* rs780094 is associated with metabolic traits including higher levels of TAG, even though glycemia levels were adequate [38]. This involvement in metabolism may interfere in the frequency of physical exercise performance.

Our results revealed that total cholesterol levels are significantly different concerning genotype, which may lead to possible influence of this polymorphism on cholesterol metabolism. Likewise studies of other *GCKR* polymorphisms show similar outcomes [39]. Even so, it is necessary to consider that the differences observed in our study concerning lipid profile, showed significant differences depending also on gender. Although the differences observed on HDL levels between men and women are in line with the existing bibliography, minority allele carriers who presented higher levels of cholesterol, also tended to consume more dietary fat [40,41]. This fact could influence the levels of blood cholesterol so further studies are needed to be conducted in this respect.

Conversely to results found in previous researches, our analyses show no significant differences for glucose and TAG according to rs780094 *GCKR* polymorphism [38]. Understanding the regulatory mechanisms of CGK activity is a complex issue. On the one hand, GCK glucokinase activity is regulated by fructose 6-phosphate (F6P), and fructose 1-phosphate (F1P) whose presence enhances and reduces GKRP-mediated inhibition, respectively [42]. On the other hand, several studies suggested that nuclear interaction with GKRP plays an important role in establishing and regulating GCK protein concentration, a fact that is essential for the maintenance of glucose homeostasis [42]. 

Regarding physical exercise practice and genetic association, it has been found that around 40% of common homozygous genotype (CC) carriers do not perform any type of exercise per week, compared to 86% of homozygous variants (TT) that perform at least some physical exercise once a week. Involvement of GCKR in lipid and glucose metabolism, support the idea about the existence of an influence on the capability to perform more frequently physical exercise. However, according to Alfred et al. (2013) no evidence about cognitive and physical capability has been found at this moment concerning this genotype [43]. 

In this study we used the exercise classification variable as a measure of the exercise practice of the individuals. There are alternatives, and complementary, measures of physical activity like the Minnesota questionnaire. In fact, we observed a significant correlation of exercise variable and energy expenditure by this questionnaire (see above), although we did not find a significant association between the rs780094 SNP and the energy expenditure (Table 1). This could be due to the Minnesota questionnaire being a more comprehensive measure of «total» physical activity during the day, while our exercise variable would be more focused on physical exercise or sports practice, that is, extra physical activity performed after work hours. In this way, the rs780094 SNP could be associated to this fraction of the total physical exercise of the individual. 

Although the genetic factor seems to be associated with a predisposal to a greater practice of physical exercise, it is crucial to take into account other multiple factors that may condition this attitude. These factors may depend on aptitudes, preferences, incentives or other aspects that might influence the level of difficulties of the physical activities performed [44]. Therefore, it is important to implement new studies to determine the genetic load on other possible factors involved in the frequency of physical exercise practice. A possible predisposition must be always validated in studies where all the conditioning variables are included. Regarding this, it is necessary to note that in the current study, all factors associated with this possible behavior have not been taken into account. Therefore, on the basis of this consideration, this study could be considered a starting point for developing future studies addressing this issue by including in the criteria these factors.

Since *GCKR* is a gene involved in the regulation of carbohydrates and lipids metabolism, it would be reasonable to assume that this gene may be also implicated in the genetic disposition to perform more frequently physical exercise. Among genes associated with adherence to physical activity practice we might also encounter the dopamine receptor 1 (*DRD1*), as well as the helix-loop-helix 2 (*NHLH2*), which are involved in eating behavior [45]. In addition, genetic variants of the *MC4R* and *LEPR* genes have also been shown to be associated with levels of physical activity according to their genotype [45]. The *GCKR* gene, is implicated in coding molecules involved in lipid and glucose turnover which refer to the fundamental pathways in body homeostasis during the practice of physical exercise.

GCK enzyme in humans is involved as a “glucose sensor” in liver, that permits to regulate the changes in plasma glucose concentrations [46]. GCK activity is potently controlled by GCKR, which is encoded by the *GCKR* gene. During fasting, GKRP is bound to GCK inhibiting its activity and locating in the nucleus. In this way, glycogenolysis and gluconeogenesis take place, and glucose is exported to the circulation for use by peripheral tissues. In postprandial state, GKRP releases GCK by glucose and F1P. GCK then binds to glucose, adopt the closed (catalytically active) conformation and exit from the nucleus to generate glucose-6-phosphate for triglycerides and glucose disposal and storage (glycogen synthesis). Alterations of the *GCKR* gene cause high levels of regulation of the metabolism of lipids and glucose and, this could determine a different frequency of physical exercise predisposition. In the case of the T/T genotype (rs780094 *GCKR*), it has been reported that insulin and blood glucose levels could remain better regulated [38], which may improve the use of substrates during physical exercise. In turn, increased levels of triglycerides in blood may favor the use of this substrate during exercise practice, and therefore preventing obesity.

Interestingly, results from this study revealed that minor genotype (TT) carriers have a higher fat consumption and tend to have greater energy intake although they are also more predisposed to perform more exercise. These outcomes would explain why anthropometric parameters were no modified, although their cholesterol levels are significantly higher. In this context, a greater motivation to perform more physical exercise, for major genotype carriers (majority of the European population), could benefit them. In the case of rare genotype carriers, recommendations could be aimed at monitoring blood cholesterol levels based on their fat intake.

## 5. Conclusions

The analysis of *GCKR* rs780094 variant may be useful to feature those subjects less prone to physical exercise. Thus, common allele carriers could benefit from personalized intervention strategies that would consider increasing the frequency of physical exercise. Likewise, this knowledge could contribute to prevent and manage overweight and obesity in the subjects with lower frequency of physical exercise. In the same way, in the case of minor genotype carriers, recommendations could be aimed at monitoring blood cholesterol levels based on their fat intake. Nevertheless, future studies will be needed to confirm these findings since the use of genetic information for the identification of individuals at risk of a given condition requires replication and a complete validation process.

## Figures and Tables

**Figure 1 genes-10-00570-f001:**
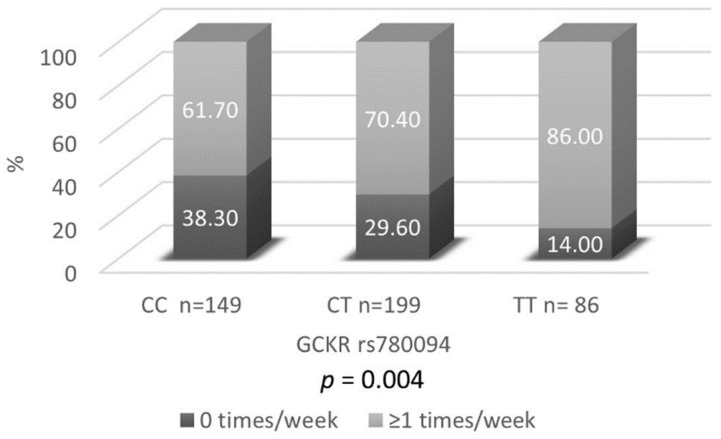
Percentage of the population classified according to the number of times that they perform physical exercise according to the genotype. *p*-Value of an additive model adjusted by 64 SNPs is also shown.

**Table 1 genes-10-00570-t001:** Descriptive characteristics of the study participants (Mean, SD).

Ariables	Gender			Exercise Classification			Genotype (rs780094)			
	Male *n* = 155	Female *n* = 402	*p*	No Activity *n* = 150	Activity *n* = 340	*p*	Common Homozygous (C/C) *n* = 155	Heterozygous (C/T) *n* = 204	Rare Homozygous (T/T) *n* = 92	*p*
Nutritional status										
Body Weight (kg)	84.29 (14.70)	67.69 (13.45)	<0.001	73.83 (17.63)	71.23 (14.64)	0.117	73.39 (16.73)	71.99 (15.27)	70.51 (14.34)	0.375
Height (cm)	176.26 (6.44)	162.30 (6.36)	<0.001	164.69 (8.19)	166.52 (9.11)	0.029	166.87 (9.45)	165.77 (8.51)	164.54 (8.85)	0.164
BMI (kg/m^2^)	27.18 (4.78)	25.72 (4.84)	0.003	27.09 (5.45)	25.65 (4.54)	0.005	26.26 (5.06)	26.18 (4.91)	26.04 (4.76)	0.949
Fat mass (%)	24.87 (7.96)	36.66 (8.34)	<0.001	35.74 (9.45)	32.66 (9.73)	0.001	32.93 (10.24)	34.01 (9.53)	34.38 (10.45)	0.495
Lean mass (%)	35.76 (4.93)	26.57 (3.58)	<0.001	27.92 (4.87)	29.39 (5.92)	0.004	29.61 (5.95)	28.55 (5.59)	28.62 (5.97)	0.214
Visceral fat classification	9.75 (5.30)	6.32 (2.80)	<0.001	7.78 (4.59)	6.93 (3.58)	0.049	7.27 (4.04)	7.26 (3.98)	6.93 (3.17)	0.715
Waist circumference (cm)	94.39 (14.07)	84.41 (13.71)	<0.001	89.12 (15.76)	86.05 (13.80)	0.042	87.07 (14.49)	87.51 (14.81)	85.98 (13.39)	0.695
Waist-Hip ratio	0.89 (0.08)	0.81 (0.08)	<0.001	0.83 (0.10)	0.82 (0.09)	0.283	0.82 (0.08)	0.83 (0.09)	0.82 (0.08)	0.493
Biochemical parameters										
Glucose (mg/dL)	88.58 (16.94)	86.77 (10.92)	0.358	88.00 (16.38)	87.76 (10.36)	0.891	86.94 (10.37)	88.92 (14.23)	89.37 (12.56)	0.311
Total-C (mg/dL)	194.77 (38.67)	202.75 (35.42)	0.048	199.06 (36.05)	200.81 (37.74)	0.659	194.93 (38.07)	205.79 (35.93)	203.17 (35.73)	0.041
HDL-C (mg/dL)	46.57 (10.84)	58.34 (14.04)	<0.001	55.65 (15.59)	55.83 (13.79)	0.912	54.59 (15.05)	57.44 (14.06)	55.00 (12.05)	0.185
LDL-C (mg/dL)	126.77 (32.68)	126.41 (29.87)	0.916	125.55 (30.67)	126.85 (31.19)	0.704	122.37 (29.96)	128.84 (31.27)	127.98 (30.70)	0.185
TAG (mg/dL)	103.39 (50.34)	95.53 (46.27)	0.138	99.06 (46.20)	96.28 (47.60)	0.585	90.92 (39.11)	98.95 (50.78)	102.33 (53.76)	0.151
TyG Index	4.56 (0.24)	4.48 (0.23)	0.009	4.51 (0.23)	4.49 (0.23)	0,543	4.46 (0.21)	4.51 (0.25)	4.54 (0.24)	0.072
Total-C/HDL-C	4.43 (1.30)	3.69 (1.03)	<0.001	3.86 (1.11)	3.83 (1.14)	0.808	3.83 (1.20)	3.81 (1.14)	3.82 (0.90)	0.988
LDL-C/HDL-C	2.87 (0.99)	2.3 (0.81)	<0.001	2.46 (0.92)	2.41 (0.86)	0.629	2.41 (0.88)	2.40 (0.89)	2.43 (0.78)	0.962
Dietary intake										
Energy (TCV: kcal/day)	2325 (650)	2037 (719)	<0.001	2121 (842)	2150 (674)	0.714	2176 (809)	2049 (580)	2260 (929)	0.074
CHO (TCV%)	37.79 (6.99)	38.48 (6.30)	0.311	38.42 (6.51)	38.34 (6.42)	0.911	39.03 (6.16)	38.30 (6.36)	37.13 (6.91)	0.108
Proteins (TCV%)	17.23 (3.27)	17.28 (3.51)	0.894	17.22 (3.29)	17.14 (3.41)	0.808	17.01 (3.12)	17.38 (3.54)	17.04 (3.30)	0.549
Fats (TCV%)	39.98 (6.38)	40.04 (6.38)	0.927	40.59 (5.99)	39.72 (6.50)	0.163	39.56 (5.92)	39.74 (6.35)	41.52 (5.90)	0.036
Water intake (mL)	1495 (622)	1454 (727)	0.522	1320 (738)	1568 (694)	0.001	1544 (798)	1450 (667)	1476 (687)	0.511
Physical activity performance										
Exercise (times/week)	2.34 (1.67)	1.73 (1.50)	<0.001	0.00 (0.00)	2.72 (1.13)	N.A.	1.79 (1.69)	1.85 (1.53)	2.31 (1.37)	0.016
Energy expenditure physical activity/week (kcal)	2701 (2434)	2038 (1676)	0.007	1504 (1656)	2533 (1948)	<0.001	2222 (1935)	2199 (1899)	1975 (1811)	0.582

BMI, Body mass index; CHO, carbohydrates; HDL-C, high density lipoprotein cholesterol; LDL-C, low density lipoprotein; N.A., not applicated; TyG Index, Triglycerides/glucose index; SD, standard deviation; TCV, total caloric value, Total-C, total cholesterol. No activity, 0 times of physical activity performance per week. Activity, one or more times of physical activity performance per week. *p*-values were obtained from one-way ANOVA for continuous variables, and Fisher exact test for categorical variables. Significance level *p* ≤ 0.05.

**Table 2 genes-10-00570-t002:** Association of exercise classification performance and genotype according to genetic models.

Model	No Activity			Activity			OR (CI)	*p*-Value ^1^
	Major Allele Homozygote	Heterozygote	Minor Allele Homozygote	Major Allele Homozygote	Heterozygote	Minor Allele Homozygote		
	*n* (%)	*n* (%)	*n* (%)	*n* (%)	*n* (%)	*n* (%)		
Codominant	57 (38.30)	59 (29.60)	12 (14.00)	92 (61.70)	140 (70.40)	74 (86.00)	1.52 (0.96–2.41)/4.02 (2.04–8.51)	0.012
Additive	57 (38.30)	59 (29.60)	12 (14.00)	92 (61.70)	140 (70.40)	74 (86.00)	1.86 (1.36–2.56)	0.004
Dominant	57 (38.30)	71 (24.90)		92 (61.70)	214 (75.10)		3.18 (1.70–6.46)	0.012

^1^ Adjusted by sex, age and BMI. *p*-value was corrected by multiple SNP comparisons (Bonferroni). Significance level *p* ≤ 0.05. No activity, 0 times of physical activity performance per week. Activity, 1 or more times of physical activity performance per week.

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
