# Peer review of "GCKR* rs780094 Polymorphism as A Genetic Variant Involved in Physical Exercise"

_genes, 2019, doi:10.3390/genes10080570_

Round 1
Reviewer 1 Report
The study presents the glucokinase regulatory protein (GCKR) gene; which has a scarcity of knowledge about the mechanisms that trigger the practice of physical exercise and energy expenditure. Besides no significant differences for glucose and TAG according to rs780094 GCKR polymorphism, is a preliminary, evaluation work that can still be increased. A positive point was the association between genotype and physical exercise phenotype. This type of analysis may be the starting point for future work and should be encouraged, verifying the possible predisposition. The results show a proper evaluation of the data. The negative point was the lack of a treadmill exercise protocol, with maximum VO2 assessment, applied to the subjects, being able to regulate and evaluate the execution and energy expenditure. The Minnesota questionnaire being a more comprehensive measure of «total» physical activity was discussed. It would be possible the geographic location of the group evaluated, and it is not clear the ethnicity or the origin of the subjects?
Author Response
We would like to thank you for your helpful comments with regard to the submission of our paper entitled "GCKR rs780094 polymorphism as a genetic variant involved in physical exercise".
Point 1. The study presents the glucokinase regulatory protein (GCKR) gene; which has a scarcity of knowledge about the mechanisms that trigger the practice of physical exercise and energy expenditure. Besides no significant differences for glucose and TAG according to rs780094 GCKR polymorphism, is a preliminary, evaluation work that can still be increased. A positive point was the association between genotype and physical exercise phenotype. This type of analysis may be the starting point for future work and should be encouraged, verifying the possible predisposition. The results show a proper evaluation of the data. The negative point was the lack of a treadmill exercise protocol, with maximum VO2 assessment, applied to the subjects, being able to regulate and evaluate the execution and energy expenditure. The Minnesota questionnaire being a more comprehensive measure of «total» physical activity was discussed. It would be possible the geographic location of the group evaluated, and it is not clear the ethnicity or the origin of the subjects?
Response 1: Thank you for the proposal. As suggested, the following sentence has been added in the Materials and Methods section for clarification (page 2-3, lines 95-96):
“Participants were recruited in Cantoblanco Campus (Autonomous University of Madrid. Madrid, Spain), being almost entirely of Caucasian origin.”
Reviewer 2 Report
The authors present an exploratory study on the effect of 64 genetic variants related to inflammation, circadian rhythms, vascular function and energy, lipid and carbohydrate metabolism, on dietary intake, energy expenditure and the time the individuals performed physical exercise in 451 participants aged from 18 to 65 years. The topic is very interesting and the article is well written. However, two points need to be addressed in order to improve the quality of the paper.
(1) Introduction, page 2, lines 65-67: The authors claim that the GCKR gene might interfere with the amount of physical exercise performed. It would be important for the reader to clarify the hypothetical mechanism leading to a sedentary behavior for CC homozygous participants and a significantly higher level of physical exercise for TT homozygous particpants. When I use the term 'mechanism', I think about a cascade of physiological as well as psychological phenomena leading to the decision making of practicing physical exercise or not.
(2) Method, page 4, lines 158-159: The authors explain that they carried out three logistic genetic models, namely additive, dominant and codominant. The authors are invited to explain in few words the difference between these three models for the good understanding of a reader not expert in statistics.
Author Response
We would like to thank you for your helpful comments with regard to the submission of our paper entitled "GCKR rs780094 polymorphism as a genetic variant involved in physical exercise".
Point 1. Introduction, page 2, lines 65-67: The authors claim that the GCKR gene might interfere with the amount of physical exercise performed. It would be important for the reader to clarify the hypothetical mechanism leading to a sedentary behavior for CC homozygous participants and a significantly higher level of physical exercise for TT homozygous participants. When I use the term 'mechanism', I think about a cascade of physiological as well as psychological phenomena leading to the decision making of practicing physical exercise or not.
Response 1: Taking into consideration this observation, it has been added in the Introduction (page 2, lines 71-73):
“Thus, a hypothetical mechanism behind a physically active or sedentary behaviour may lie on the greater or lesser availability of energy substrates, as well as the signals these substrates exert at the brain level”.
Point 2. Method, page 4, lines 158-159: The authors explain that they carried out three logistic genetic models, namely additive, dominant and codominant. The authors are invited to explain in few words the difference between these three models for the good understanding of a reader not expert in statistics.
Response 2: To clarify this aspect the following text has been included in the Materials and Methods (page 4, lines 164-168):
“These genetic models refer to the dominant or recessive character of each of the alleles. Thus, the additive model indicates a multiplicative effect depending on the number of rare alleles. In the dominant model, the less frequent allele is considered dominant (common homozygous vs heterozygous + rare homozygous), and the codominant model considers each genotype individually, with independent genetic effects”.